# Brain state and cortical layer-specific mechanisms underlying perception at threshold

**Mitchell P Morton[1,2], Sachira Denagamage[1,2], Isabel J Blume[1], John H Reynolds[3], Monika P Jadi[1,2,4,5], Anirvan S Nandy[1,2,5,6,7]***

[1]Department of Neuroscience, Yale University, New Haven, United States; [2]Interdepartmental Neuroscience Program, Yale University, New Haven, United States; [3]Systems Neurobiology Laboratories, The Salk Institute for Biological Studies, La Jolla, United States; [4]Department of Psychiatry, Yale University, New Haven, United States; [5]Wu Tsai Institute, Yale University, New Haven, United States; [6]Department of Psychology, Yale University, New Haven, United States; [7]Kavli Institute for Neuroscience, Yale University, New Haven, United States

*For correspondence:
anirvan.nandy@yale.edu

Competing interest: The authors declare that no competing interests exist.

## eLife Assessment

This **useful** study by Nandy and colleagues examined relationships between behavioral state, neural activity in cortical area V4, and trial-by-trial variability in the ability to detect weak visual stimuli. They present **solid** evidence indicating that certain changes in arousal and eye-position stability, along with patterns of synchrony in the activity of neurons in different layers of V4, can show modest correspondences to changes in the ability to correctly detect a stimulus. These findings are likely to be of interest to those who seek a deeper understanding of circuit mechanisms that underlie perception.

**Abstract** Identical stimuli can be perceived or go unnoticed across successive presentations, producing divergent behavioral outcomes despite similarities in sensory input. We sought to understand how fluctuations in behavioral state and cortical layer and cell class-specific neural activity underlie this perceptual variability. We analyzed physiological measurements of state and laminar electrophysiological activity in visual area V4 while monkeys were rewarded for correctly reporting a stimulus change at perceptual threshold. Hit trials were characterized by a behavioral state with heightened arousal, greater eye position stability, and enhanced decoding performance of stimulus identity from neural activity. Target stimuli evoked stronger responses in V4 in hit trials, and excitatory neurons in the superficial layers, the primary feed-forward output of the cortical column, exhibited lower variability. Feed-forward interlaminar population correlations were stronger on hits. Hit trials were further characterized by greater synchrony between the output layers of the cortex during spontaneous activity, while the stimulus-evoked period showed elevated synchrony in the feed-forward pathway. Taken together, these results suggest that a state of elevated arousal and stable retinal images allow enhanced processing of sensory stimuli, which contributes to hits at perceptual threshold.

## Introduction

Physical properties of stimuli strongly influence perception such that low-intensity stimuli are detected infrequently. As intensity increases, detection probability remains low until some perceptual threshold

is crossed, after which stimuli are perceived robustly. A psychometric function (*Prins and Kingdom, 2018*; *Watson, 1979*; *Wichmann and Hill, 2001*) mathematically describes this property of perception. Only within a narrow range around the perceptual threshold do stimuli lead to significant trial-to-trial perceptual variance. While many studies present stimuli at threshold (*Herman et al., 2019*; *Levitt, 1971*; *Pins and Ffytche, 2003*; *Ress and Heeger, 2003*), few have probed the laminar cortical microcircuit mechanisms that underlie successful or unsuccessful perception under these conditions (*McCormick et al., 2020*; *van Vugt et al., 2018*).

Prior studies have characterized how perceived stimuli trigger stronger information propagation from earlier visual areas to higher-order visual and frontal regions (*Herman et al., 2019*; *van Vugt et al., 2018*). This information propagation and sensory processing are strongly influenced by brain states such as arousal and attention (*McCormick et al., 2020*; *Harris and Thiele, 2011*). Arousal has long been recognized for its role in modulating cortical activity (*Livingstone and Hubel, 1981*; *McCormick and Bal, 1997*; *Vinck et al., 2015*) and affecting performance in various sensory tasks (*Aston-Jones and Cohen, 2005*; *McGinley et al., 2015*; *Yerkes and Dodson, 1908*). In visual area V4, a key intermediate region in the ventral visual processing stream (*Goodale and Milner, 1992*; *Mountcastle, 1997*; *Roe et al., 2012*), attention strongly modulates neural activity (*Desimone and Duncan, 1995*; *McAdams and Maunsell, 1999*; *Moran and Desimone, 1985*; *Reynolds et al., 2000*). Attention increases the firing rates of V4 neurons, enhances the reliability of individual neuron firing, and reduces correlated fluctuations among pairs of neurons (*McAdams and Maunsell, 1999*; *Cohen and Maunsell, 2009*; *Mitchell et al., 2007*; *Mitchell et al., 2009*). Brain state dynamics impact both cortical and subcortical structures, contributing to behavior (*Ghosh and Maunsell, 2021*; *Zénon and Krauzlis, 2012*). Fluctuations in attention are reflected in the on- and off-state dynamics of V4 ensembles, which have been shown to correlate with behavioral performance (*Engel et al., 2016*; *van Kempen et al., 2021*).

The visual cortex has a columnar architecture in which multiple cell classes (*Connors and Gutnick, 1990*; *Markram et al., 2004*; *Migliore and Shepherd, 2005*; *Wonders and Anderson, 2006*; *Zeng and Sanes, 2017*) across the cortical layers (*Mountcastle, 1997*; *Douglas and Martin, 2004*) form distinct sub-populations. These sub-populations have unique and stereotyped patterns of connectivity, thus forming a canonical microcircuit that orchestrates the encoding and flow of information (*Douglas and Martin, 2007*; *Hirsch and Martinez, 2006*). Moreover, these sub-populations contribute uniquely to sensory processing and are differentially modulated by brain states (*Mitchell et al., 2007*; *Nandy et al., 2017*; *McCormick et al., 1985*; *Pettine et al., 2019*). While it has been shown that attentional modulation varies across cortical layers (*Nandy et al., 2017*; *Pettine et al., 2019*; *Mehta et al., 2000a*; *Mehta et al., 2000b*; *Buffalo et al., 2011*; *Ferro et al., 2021*; *Westerberg et al., 2021*; *Westerberg et al., 2022*), the role of these sub-populations in attentive perception at threshold remains poorly understood. Moreover, the influence of physiological states, which may be responsible for different outcomes at threshold, on these sub-populations has not been studied in detail.

Here, we examine the neural mechanisms that regulate perception at threshold. We specifically focus on the columnar microcircuit mechanisms within area V4. We hypothesized that minor fluctuations in behavioral state, such as arousal and visual sensitivity, and in the activity of neural sub-populations across the layers of the visual cortex, result in different perceptual outcomes at threshold. Specifically, we hypothesized that output layer (II–III, V–VI) sub-populations, ones projecting to higher cortical areas and subcortical structures, would show evidence of improved capacity for stimulus representation during successful perception. We also hypothesized that such successful events would be accompanied by improved information propagation throughout a cortical column. We find that differences in behavioral states and lamina-specific neural states characterize correct and incorrect trials at threshold and explain perceptual variability.

## Results

To study the neural dynamics responsible for determining whether a stimulus presented at perceptual threshold is perceived, we analyzed behavioral and cortical layer-specific neural data from area V4, collected while monkeys performed a cued attention task (*Nandy et al., 2017*). Monkeys were trained to detect an orientation change in one of two Gabor stimuli that were presented concurrently at two spatial locations, and to report having seen the change by making an eye movement to the changed stimulus. Prior to a block of trials, monkeys were cued as to which of the two spatial locations was

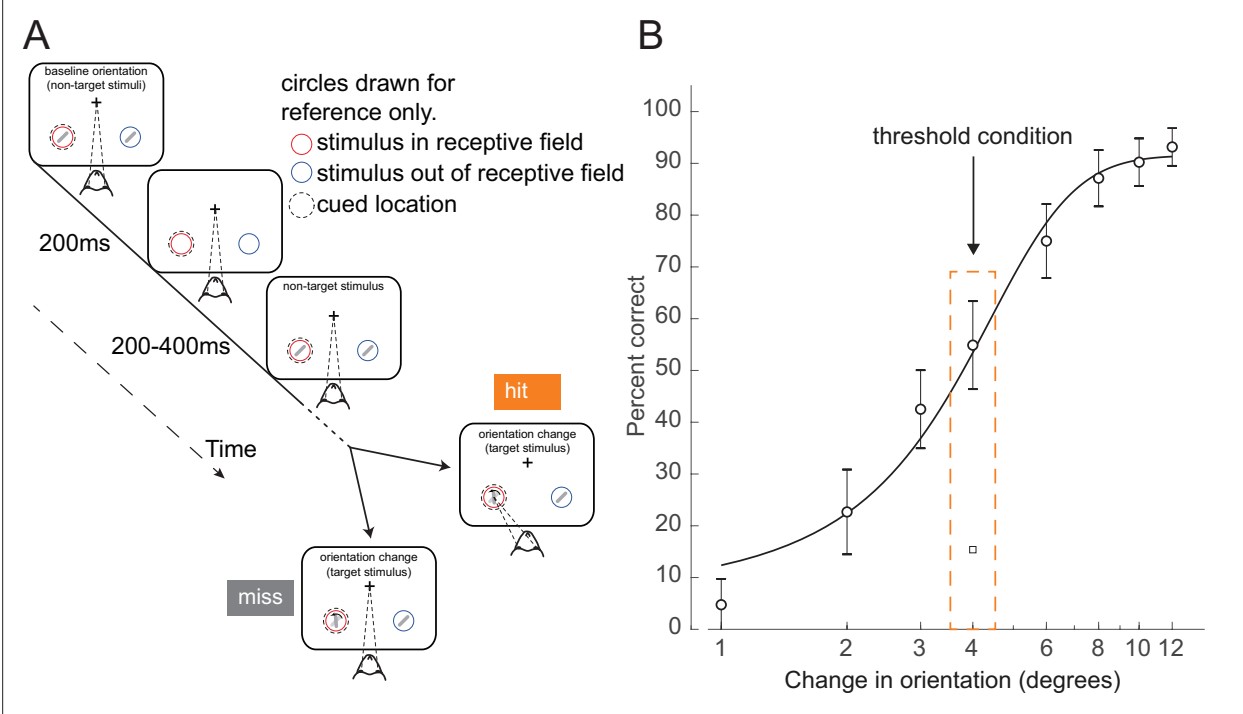

**Figure 1.** Orientation change detection task at perceptual threshold. (**A**) Schematic of task structure. The monkey initiated a trial by fixating on the center of the screen. Two Gabor stimuli (represented by oriented lines) were presented for 200 ms and then turned off for 200–400 ms. This was repeated until, at an unpredictable time, one of the stimuli changed orientation. The monkey could report having seen the change by making an eye movement to the location of the target stimulus to receive a reward (hit trials). If the monkey failed to report the orientation change and maintained fixation on the center of the screen it was not rewarded (miss trials). Before a block of trials, the monkey was cued as to which stimulus was likely to undergo the change (95% valid cue). In 5% of trials the orientation change occurred at the other location (foil trials). Circles indicating the cued location and receptive field are drawn for figure reference only and were not presented during the task. (**B**) Example behavioral psychometric function from one recording session and attention condition. Behavioral performance (hit rate, circles) is presented as a function of orientation change. Data was fitted with a logistic function. The threshold condition, trials with performance halfway between the upper and lower asymptotes of the logistic function, is indicated by the orange box. Error bars represent standard deviation calculated with a jackknife procedure (20 jackknives). The square symbol indicates foil trial performance.

The online version of this article includes the following figure supplement(s) for figure 1:

**Figure supplement 1.** Laminar recordings in V4 (modified from *Nandy et al., 2017*).

**Figure supplement 2.** Additional psychometric function examples.

**Figure supplement 3.** Psychometric function parameters.

likely to undergo the orientation change (95% valid cue; presented at the start of each block). During a trial, 'non-target' stimuli at a fixed reference orientation were repeatedly presented. Non-targets were turned on for 200 ms at the two spatial locations, and then turned off for a variable interval (200–400 ms). At a random time (1–5 s, mean 3 s) a 'target' stimulus, differing in orientation from the non-targets, was presented at one of the locations. If the monkey reported having detected the orientation change by making an eye movement to the location of the target stimulus, it received a juice reward (*Figure 1A*, 'hit' trial). If the monkey failed to detect the orientation change and instead continued to maintain fixation on the center of the monitor, it was not rewarded (*Figure 1A*, 'miss' trial). In this study, we focused exclusively on trials in which the target stimulus was presented at the cued location (95% of trials). All figures relate exclusively to trials in which the change occurred at the cued location.

On each trial, the magnitude of the orientation change was drawn from a distribution that spanned multiple levels of difficulty. We fit the behavioral data with a logistic function (*Prins and Kingdom, 2018*) and defined the threshold condition as the orientation change that was closest to the 50% threshold of the fitted psychometric function for that session (*Figure 1B*, Materials and methods; see *Figure 1—figure supplement 2* for additional examples and *Figure 1—figure supplement 3A–G*

for logistic fit parameters). We selected this subset of trials for further analysis, since the constant target stimuli in these trials were equally likely to be perceived or not perceived. Target presentation times were not different between hit and miss trials (*Figure 1—figure supplement 3H*; p=0.15, Wilcoxon rank sum test). There was a slight difference in threshold trial performance based on time in the session (*Figure 1—figure supplement 3I*, p<0.01, permutation test). Performance in the middle of the recording session (second and third quartiles) was higher than in the beginning and end of the session (first and fourth quartiles). Monkeys initiated a median of 905 trials per session (range: 651–1086).

While monkeys performed this task, single- and multi-unit activity and local field potentials (LFPs) were recorded in area V4 using 16-channel linear array electrodes (Plexon Inc, *Figure 1—figure supplement 1A–E*). The array was inserted perpendicular to the cortical surface and spanned the cortical layers. We used current source density (CSD) analysis (*Mitzdorf, 1985*) to estimate the boundaries between the superficial (I–III), input (IV), and deep (V–VI) cortical layers (*Figure 1—figure supplement 1E and F*), and assign individual neurons their layer identity (*Nandy et al., 2017*). Single units were classified as either broad-spiking (putative excitatory neurons) or narrow-spiking (putative inhibitory neurons) on the basis of their waveform width using previously published techniques (peak-to-trough duration; *Figure 1—figure supplement 1D*; see Materials and methods; *Connors and Gutnick, 1990*; *Nandy et al., 2017*; *McCormick et al., 1985*; *Kawaguchi, 1993*; *Nowak et al., 2003*). Eye position and pupil diameter were also recorded (ISCAN ETL-200). When analyzing pupil diameter and eye position data, we considered all trials in the threshold condition in which the change occurred at the cued location, regardless of whether the cued location was within the receptive field (RF) of the recorded neurons. For all electrophysiological analyses, we only considered trials in which the cued stimulus was within the RF of the recorded neurons, and the stimulus change occurred at the cued location.

To assess the behavioral impact of variations in arousal and retinal image stability across trials at the threshold condition, we compared pupil diameter and microsaccade incidence across trial outcomes. Larger pupil diameter is thought to be a proxy for elevated alertness and arousal (*McCormick et al., 2020*; *Aston-Jones and Cohen, 2005*; *McGinley et al., 2015*; *Beatty and Lucero-Wagoner, 2000*; *Hess and Polt, 1964*; *Reimer et al., 2014*; *Tang and Higley, 2020*). We found that hit trials were associated with larger pupil diameters compared to miss trials, both before and during non-target and target stimulus presentations (*Figure 2A*). We quantified this difference in the estimation statistics framework (*Calin-Jageman and Cumming, 2019*; *Ho et al., 2019*) by comparing effect sizes and using bootstrapping to estimate uncertainty in the differences. We found that the mean of the distribution of pupil diameters associated with hit trials is greater than that associated with misses (*Figure 2B*; complementary null hypothesis testing results in *Supplementary file 1a*). Prior work has shown that the optimal state for sensory performance occurs at intermediate levels of arousal, with states of low and hyper arousal associated with decreased performance (*Aston-Jones and Cohen, 2005*; *McGinley et al., 2015*; *Yerkes and Dodson, 1908*; *Cools and D'Esposito, 2011*; *Murphy et al., 2011*; *Rajagovindan and Ding, 2011*; *Vijayraghavan et al., 2007*). In both hit and miss trials, the mean pupil diameter was close to the optimal arousal state for perceptual performance (*Figure 2C*; *McGinley et al., 2015*). The average differences in pupil diameter across hit and miss trials reflect differences within the optimal state of intermediate arousal. All results were held for individual animals (*Figure 2—figure supplement 2A–C*). Our results thus suggest that hits are more likely to occur during periods of greater arousal.

Microsaccades, small fixational eye movements of <1° in amplitude that occur during normal fixation, are associated with periods of decreased visual sensitivity due to unstable retinal images (*Dicke et al., 2008*; *Zuber and Stark, 1966*). Microsaccades have been linked to suppressed neural responses in visual areas during perceptual tasks, impairing fine visual discrimination and behavioral performance (*Beeler, 1967*; *Hafed and Krauzlis, 2010*). We grouped trials in the threshold condition based on whether a microsaccade occurred in a 400 ms window preceding the onset of the target stimulus. Most trials with a pre-target microsaccade were misses, whereas the majority of trials without a microsaccade in this window were hits (*Figure 2D*; see *Figure 2—figure supplement 2D* for individual animal plots). There is a strong link between microsaccade direction and attention deployment (*Lowet et al., 2018*; *Engbert and Kliegl, 2003*; *Hafed and Clark, 2002*; *Gowen et al., 2007*; *Galfano et al., 2004*; *Pastukhov and Braun, 2010*; *Yu et al., 2022*). Consistent with previous reports

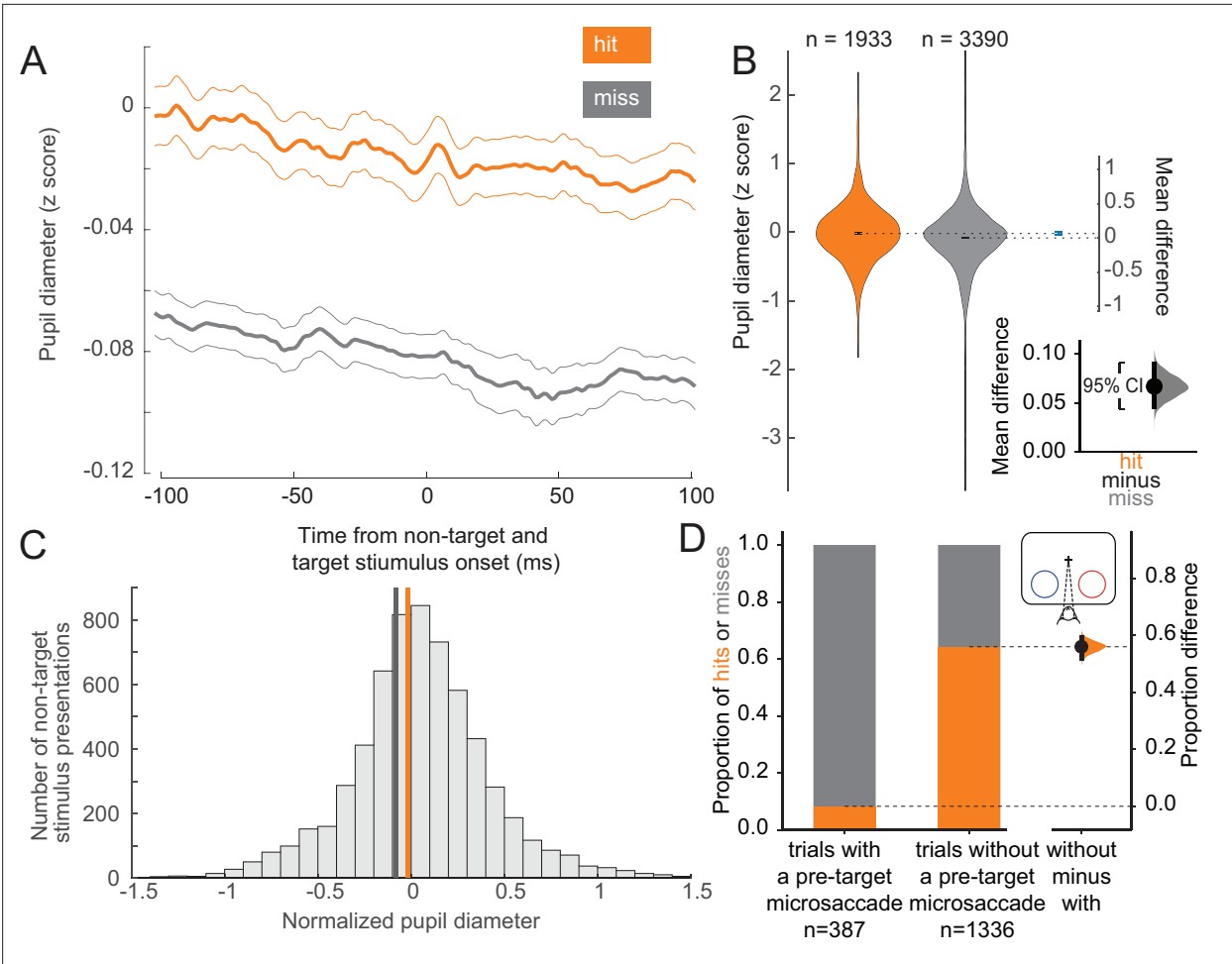

**Figure 2.** Hit trials have larger pupil diameter whereas microsaccades more often precede misses. (**A**) Normalized pupil diameter for hit and miss trials in the threshold condition. 0 ms corresponds to non-target and target stimulus onset. Mean ± s.e.m. (**B**) Distribution of pupil diameter values associated with hit and miss trials. Pupil diameter was averaged from 100 ms before to 100 ms after non-target and target stimulus onset. Violin plots were generated using kernel smoothing (see Materials and methods). Error bars represent 95% confidence intervals for the mean of each distribution, and the mean difference (blue, right axes). *Inset*: zoomed-in view of the mean difference between hit and miss trials. Black bar represents a 95% confidence interval of the mean difference. Shaded region reflects the distribution of the bootstrapped estimation of the mean difference. (**C**) Histogram of mean pupil diameter around the time of non-target and target stimulus onset (calculated as in **B**). Orange and gray lines represent the mean pupil diameter for hit and miss trials, respectively. (**D**) *Left:* Hit rate for trials with (387 trials) and without (*right*, 1336 trials) a microsaccade detected in the time window 0–400 ms before target onset. *Right:* Bootstrapped estimation of the mean difference in hit proportion in trials with vs without a pre-target microsaccade. Same conventions as in B.

The online version of this article includes the following figure supplement(s) for figure 2:

**Figure supplement 1.** Microsaccades are preferentially directed toward the target in correct trials and have a slight correlation with pupil diameter.

**Figure supplement 2.** Single monkey pupil diameter and microsaccade data.

we also find that microsaccades toward the attended stimulus were overrepresented in correct trials (*Figure 2—figure supplement 1A*, upper left). Conversely, microsaccades toward the attended stimulus were underrepresented in incorrect trials (*Figure 2—figure supplement 1A*, lower left). There was a very low but statistically significant negative correlation between pupil diameter and microsaccade rate (*Figure 2—figure supplement 1B*, $r^2 = 0.006$, $p < 0.001$). Microsaccade rates and intermicrosaccade times are reported in *Figure 2—figure supplement 1C and D*. Overall, these results suggest that successful trials at threshold are significantly more likely to occur during a state of greater arousal and improved visual sensitivity.

Having established that hit trials are more likely to occur in states of elevated arousal and visual stability, we investigated whether hits are characterized by differential information processing in V4.

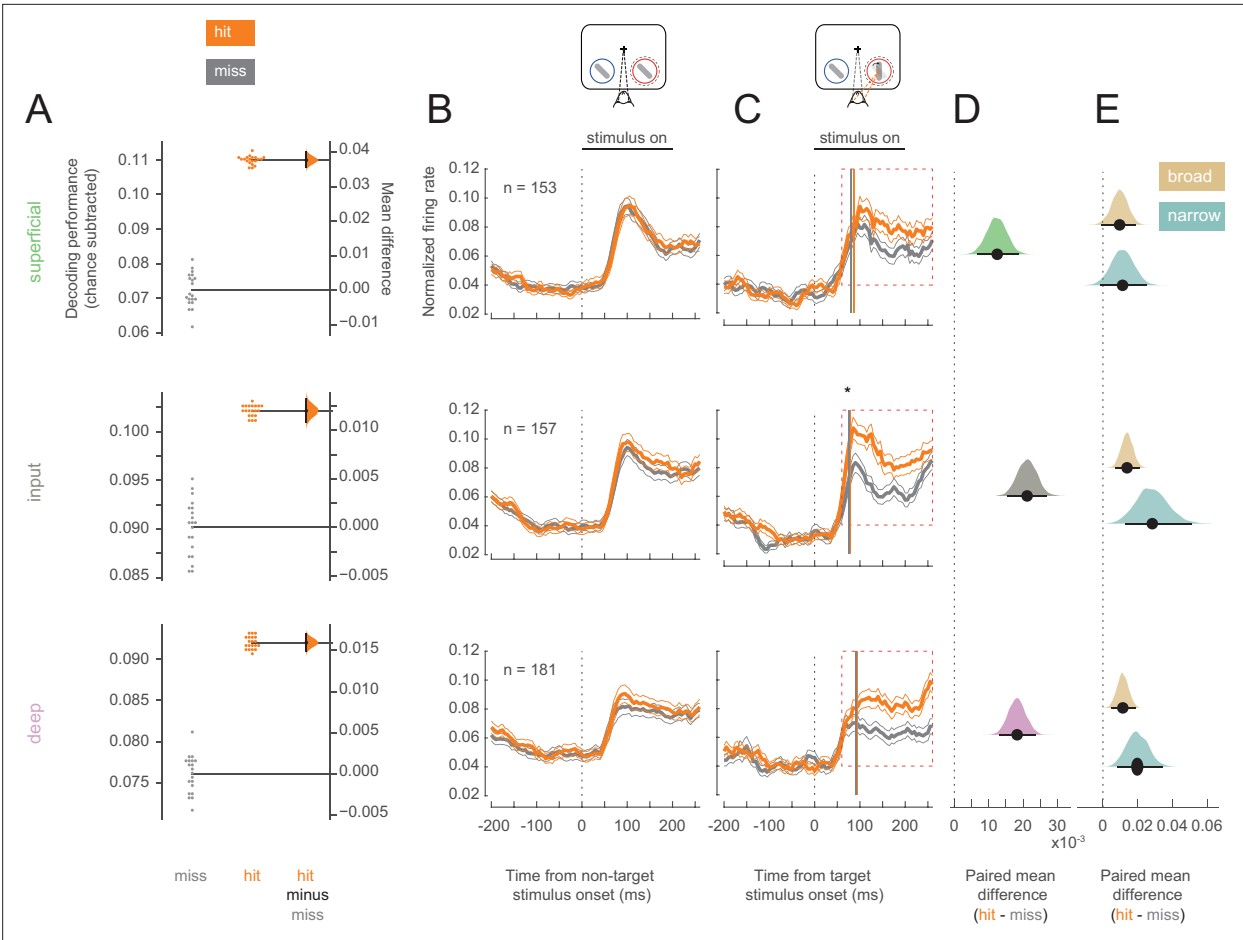

**Figure 3.** Target stimuli evoke higher firing rates in hit trials. Rows correspond to different layers (top = superficial, middle = input, bottom = deep). (**A**) Performance for decoding targets from non-targets from single units and multi-units in each layer. Points in the left section of each plot show the decoding performance for each of the 20 different cross-validations. The right section for each layer shows the bootstrapped estimation of the difference between decoding performance between hits and misses. Half-violin plots show the bootstrapped distribution of the difference, and black dots and bars represent the mean and 95% confidence intervals of the difference in decoding performance. Chance levels, determined by shuffling target and non-target identity, were subtracted from the raw decoding performance values. (**B**) Non-target population (single- and multi-unit) PSTH of visually responsive neurons for the hit (orange) and miss (dark-gray) trials in the threshold condition (mean ± s.e.m.). The horizontal black bar indicates the time and duration of stimulus presentation. (**C**) As in B but for target stimuli. The star indicates the time at which firing rates in the input layer first differ significantly between hit and miss trials. Vertical lines represent the mean time at which firing rates for each neuron rise above the 95% confidence interval of their baseline activity (see also *Figure 3—figure supplement 3C*). (**D**) Bootstrapped estimation of the paired mean difference in target stimulus-evoked firing rate between hit and miss trials in the time window 60–260 ms (red dotted box in C) after target stimulus onset. Shaded regions represent the bootstrapped estimation of the paired mean difference in firing rate (hit-miss), and black lines are 95% confidence intervals. Plots include data from both single and multi-units, separated by layer (top = superficial, middle = input, bottom = deep). (**E**) As in D, bootstrapped estimation of the paired mean difference in firing rate for hit trials compared to miss trials in the target stimulus-evoked period, but only for single units broken up by cell class (gold = broad, teal = narrow).

The online version of this article includes the following figure supplement(s) for figure 3:

**Figure supplement 1.** Single monkey decoding performance.

**Figure supplement 2.** Single monkey firing rate data.

**Figure supplement 3.** Firing rates for individual neurons and reaction time in threshold condition.

We first examined the ability to discriminate target stimuli from non-target stimuli using the firing rates of single- and multi-unit V4 neurons in each of the three identified cortical layers (superficial, input, and deep). A linear decoder could better discriminate targets from non-targets in hits compared to misses (*Figure 3A*; see *Figure 3—figure supplement 1* for individual animal plots), suggesting

differences in firing rates across these trial types. This improved stimulus discriminability was consistent across all three layers (*Figure 3A*).

Elevated stimulus-evoked firing rates would indicate a stronger representation of the stimulus that could cause this improved discriminability in hits. We compared the firing rates of all neurons (single- and multi-units) recorded in each cortical layer across hit and miss trials. For non-target stimuli, firing rates were equivalent for hits and misses in both the pre-stimulus (0–200 ms before stimulus onset) and stimulus-evoked (60–260 ms following stimulus onset) periods (*Figure 3B*; see *Figure 3—figure supplement 2* for individual animal plots). For the target stimulus, firing rates were once again equivalent in the pre-stimulus period, but hit trials were characterized by elevated firing across cortical layers in the stimulus-evoked period (*Figure 3C and D*). Broad- and narrow-spiking neurons in both the input and deep layers respond more to target stimuli in hit trials, and trend toward elevated firing rates in the superficial layers during hits (*Figure 3E*). The average firing rate in response to target stimuli for each neuron is shown in *Figure 3—figure supplement 3A* for both hit and miss trials. It is important to note that the stimuli presented to the animals were identical for both hits and misses. Moreover, the responses to the target stimuli occur early, and elevated firing in hits emerges at the time of expected

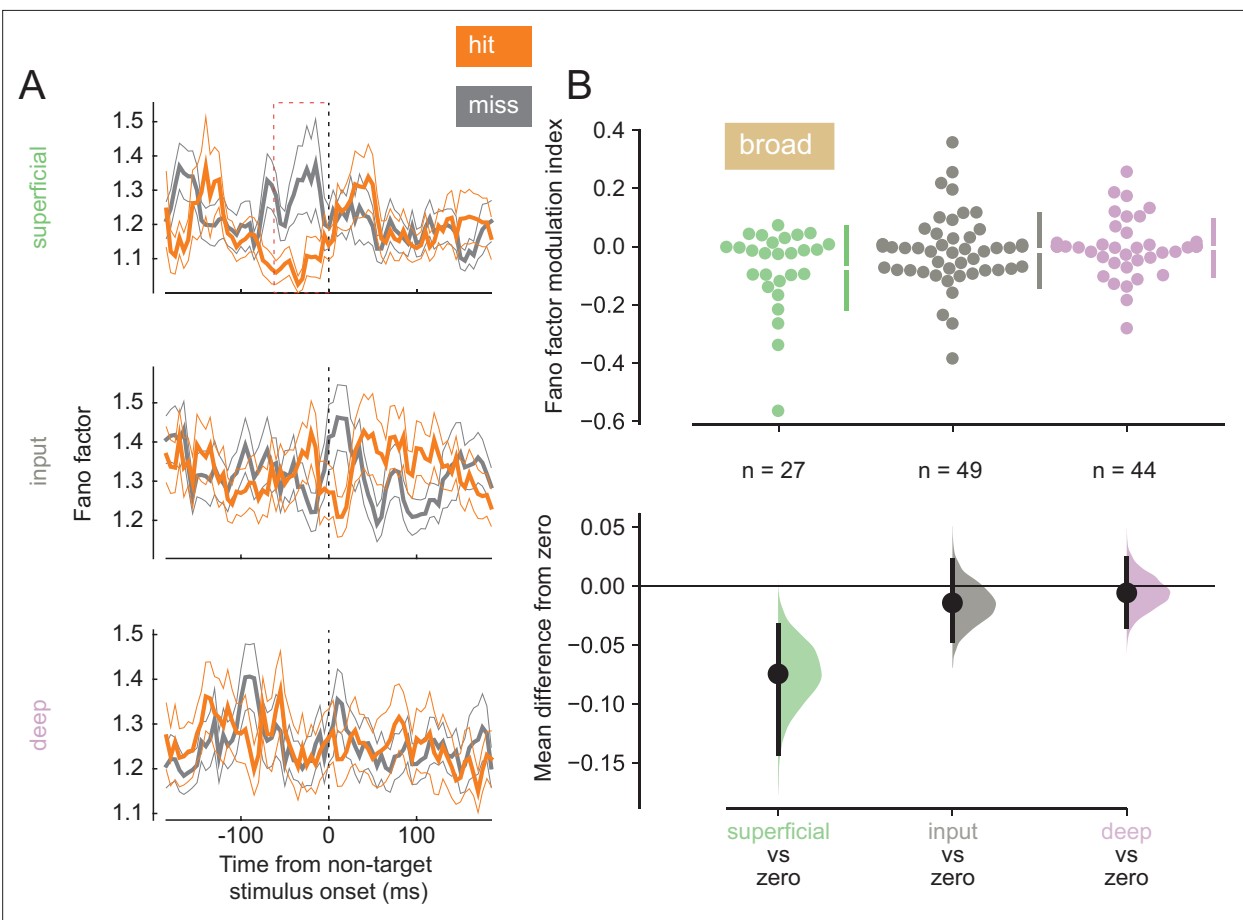

**Figure 4.** Broad-spiking neurons in the superficial layer have decreased variability in hit trials. (**A**) Rows correspond to different layers (top = superficial, middle = input, bottom = deep). The Fano factor of broad-spiking putative excitatory neurons for the hit and miss trials in the threshold condition (mean ± s.e.m.). There is a significant decrease in variability for the hit trials prior to stimulus onset only in the superficial layer. 0 ms corresponds to non-target stimulus onset. The average Fano factor within a 60 ms time window (red dashed box) prior to non-target stimulus onset is plotted in **B**. (**B**) *Top:* Fano factor modulation index for each broad-spiking neurons recorded in each layer, averaged in the 60 ms preceding non-target stimulus onset. *Bottom:* Bootstrapped estimation of the mean difference of the Fano factor modulation index from zero in each of the three layers. Colored curves represent the estimated bootstrapped distribution. Black dots and lines reflect the mean and 95% confidence intervals of the distributions.

The online version of this article includes the following figure supplement(s) for figure 4:

**Figure supplement 1.** Single monkey Fano factor data.

**Figure supplement 2.** Narrow-spiking neurons do not have decreased variability in hit trials.

V4 response latencies (70–100 ms; *Figure 3—figure supplement 3C*), and thus cannot be attributed to elevated levels of firing due to subsequent saccade planning in these trials (*Figure 3—figure supplement 3B*; expected >200 ms; *Steinmetz and Moore, 2014*).

Variability in response reflects how reliably information is encoded by a neural population. Lower baseline variability can enhance the ability of neurons to encode stimulus differences. We calculated the Fano factor, a mean-normalized measure of trial-to-trial variability in firing, for single units in our population (*Figure 4A*; see *Figure 4—figure supplement 1* for individual animal plots). We find that broad-spiking units in the superficial layer exhibited lower Fano factor during the pre-stimulus period in hit trials (0–60 ms before non-target stimulus onset, *Figure 4B*), indicating this population of neurons fired more reliably when the animal correctly detected the orientation change. This was not the case for broad-spiking neurons in other layers (*Figure 4B*) or narrow-spiking neurons (*Figure 4—figure supplement 2*).

We next wanted to test how the relationship between spiking activity and LFPs may differ across hits and misses. Spike-LFP synchrony can reflect cortical processing and both within- and inter-areal coordination (*Fries, 2009*; *Fries et al., 2008*; *Siapas et al., 2005*). We calculated the PPC (*Vinck et al., 2010*), a frequency-resolved measure of spike-LFP phase-locking, for single and multi-units relative to their local LFP signal during the pre-stimulus period (0–200 ms before non-target stimulus onset, *Figure 5A*; see *Figure 5—figure supplement 2* for individual animal plots). We averaged PPC values at low (3–12 Hz), medium (15–25 Hz), and high (30–80 Hz) frequency bands (superficial and input: *Figure 5—figure supplement 1A and B*; deep: *Figure 5B*, *Figure 5—figure supplement 1*). Deep-layer neurons exhibit reduced low-frequency phase-locking in hit trials than in misses (*Figure 5B*). This desynchronization during hits is similar to prior reports of desynchronization due to the deployment of attention (*Mitchell et al., 2009*; *Nandy et al., 2017*).

Our results at the individual neuron or neural-sub-population levels suggest enhanced processing of perceived stimuli. However, it is the concerted activity among neural sub-populations that ultimately determine information flow through the laminar cortical circuit. We turned to canonical correlation analysis (CCA) to investigate the strength of feed-forward communication across layers (*Mitra, 2007*). CCA has previously been used to describe interactions among multiple cortical areas (*Semedo et al., 2022*). We performed CCA on each pair of layers: input to superficial, input to deep, and superficial to deep, where the two elements in each pair correspond to the upstream and downstream layers respectively (*Figure 6A*). We refer to the results of CCA as population correlations. Interlaminar feed-forward population correlations were higher in hits than in misses in both the pre-stimulus and stimulus-evoked periods (*Figure 6B and C*). This suggests that feed-forward information flow through the column is more effective in hits than in misses.

To further investigate interlaminar communication, we analyzed interlaminar synchrony as signatures of differential information flow between hit and miss trials. Spike-spike coherence (SSC) is a frequency-resolved measure of the degree to which two spike trains fluctuate together (*Mitchell et al., 2009*; *Mitra and Pesaran, 1999*). We measured interlaminar SSC for spike trains from pairs of cortical layers, each spike train being comprised of all recorded action potentials in a given layer (see Materials and methods). We computed interlaminar SSC separately for hit and mis trials in both the pre-stimulus (0–200 ms before non-target stimulus onset, *Figure 7A*) and non-target stimulus-evoked (60–260 ms after non-target stimulus onset, *Figure 7C*) periods, matching the firing rates across hit and miss trials separately for the pre-stimulus and non-target stimulus-evoked conditions (see *Figure 7—figure supplement 1* for individual animal plots). We averaged SSC for each pair of layers across three frequency bands, 3–12 Hz, 15–25 Hz, and 30–80 Hz (*Figure 7B and D*).

Overall, hit trials have greater interlaminar SSC compared to misses at almost all frequencies (*Figure 7B and D*). In the pre-stimulus period, the strongest SSC difference between hits and misses was observed between the superficial and deep layers across all frequencies (*Figure 7B*, middle panel). This implies greater synchrony of the output layers of the cortex during hit trials. In contrast, this pattern was directionally the same during the non-target stimulus-evoked period, but stronger in the other layer pairs, with greater SSC differences being found in pairs that involve the input layer (*Figure 7D*, top and bottom). This may reflect a higher degree of stimulus-driven feed-forward information propagation during hit trials. When comparing across time (pre-stimulus vs non-target stimulus-evoked), layers, and frequency band, there was a significant interaction effect of layer pair and time window (three-way ANOVA, $p = 0.0075$).

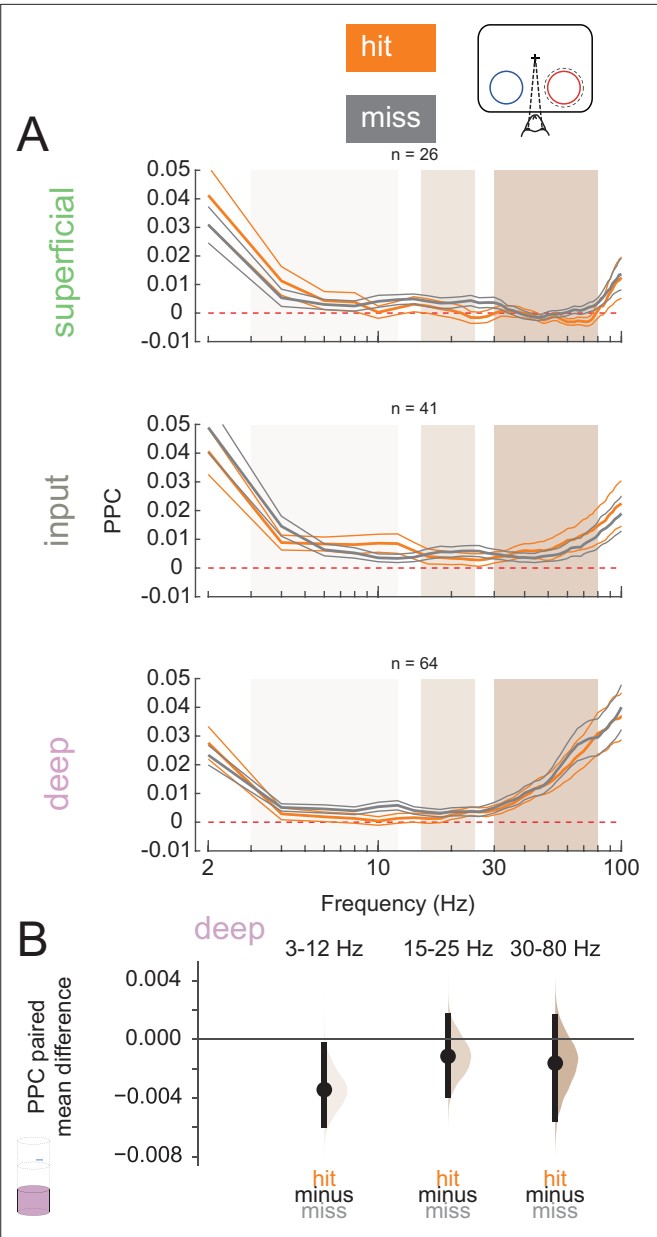

**Figure 5.** Deep-layer neurons are phase-locked to low-frequency rhythms in miss trials. (**A**) Pairwise phase consistency (PPC) of single and multi-units in each layer to the local field potential (LFP) signal recorded from the same channel in hit and miss trials at threshold. PPC was calculated in the pre-stimulus period (0–200 ms before stimulus onset). Dashed red line indicates a PPC of 0, below which there is no consistent relationship between spikes and LFP phase. (**B**) Bootstrapped estimation plot for the paired mean difference in PPC for deep-layer neurons over three frequency bands: 3–12 Hz, 15–25 Hz, 30–80 Hz. Curves represent the bootstrapped distribution for the paired difference, and black dots and vertical lines represent the mean and 95% confidence intervals for the paired mean difference.

The online version of this article includes the following figure supplement(s) for figure 5:

**Figure supplement 1.** Additional pairwise phase consistency (PPC) data.

**Figure supplement 2.** Single monkey pairwise phase consistency (PPC) data.

Finally, we sought to compare the predictive power of our results on the monkey's perceptual performance. We created a generalized linear model (GLM) to regress behavioral outcome from the pupil diameter, number of microsaccades in the pre-target window, and average target-evoked multi-unit firing rate in each of the three layers (see Materials and methods; *Davis*

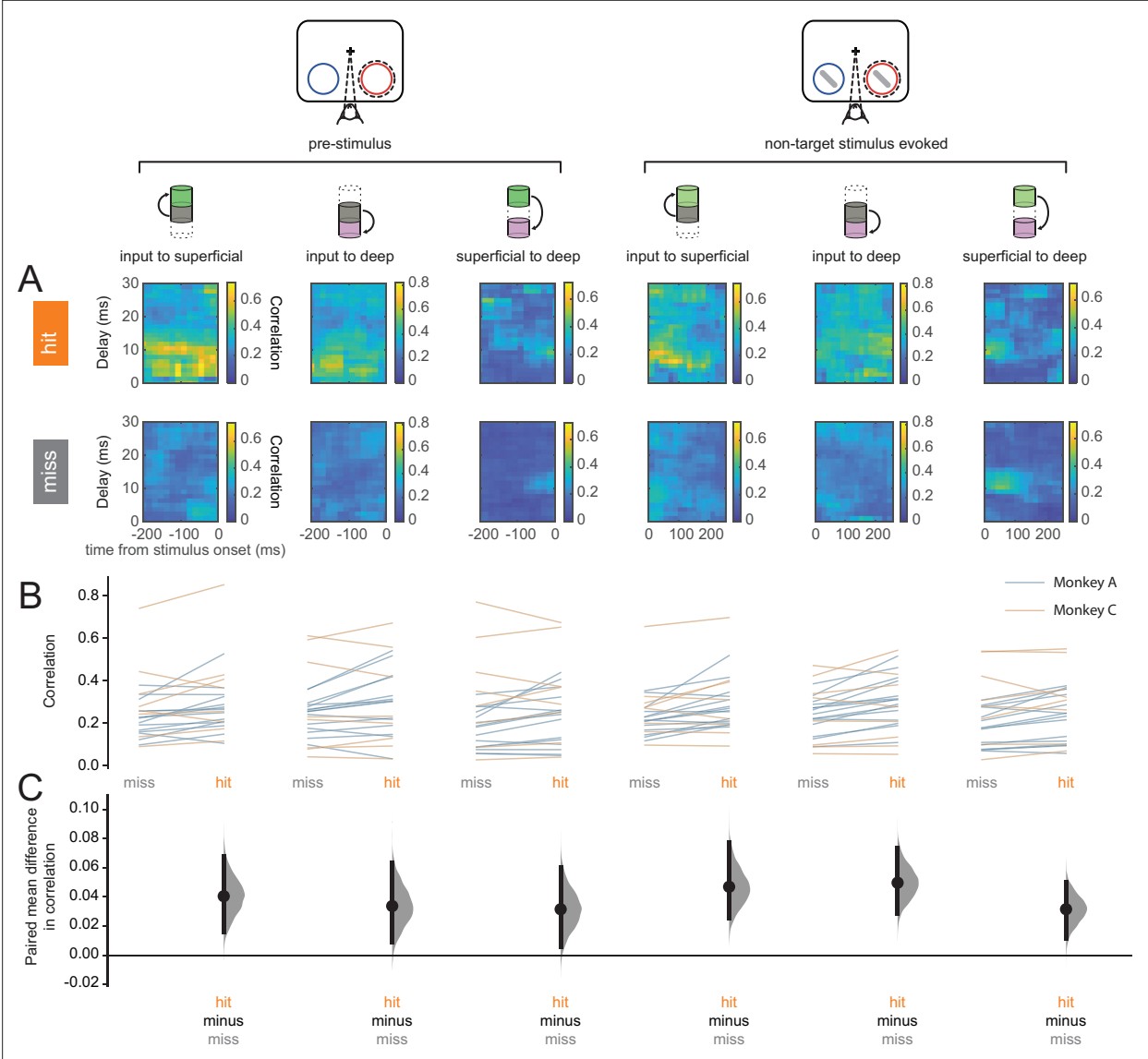

**Figure 6.** Hit trials are characterized by stronger feed-forward interlaminar population correlations. (**A**) Canonical correlation analysis (CCA)-based population correlation as a function of time and interlaminar delay during the pre-stimulus and stimulus-evoked periods in hit and miss trials in an example session. (**B**) Mean feed-forward population correlation in each session. Color indicates the monkey (blue = Monkey A, yellow = Monkey C). (**C**) Bootstrapped estimation plot for the paired mean difference in population correlation for each pair of layers and time window (pre-stimulus or stimulus-evoked). Curves represent the bootstrapped distribution for the paired difference, and black dots and vertical lines represent the mean and 95% confidence intervals for the paired mean difference.

*et al., 2020*). Other reported measures (Fano factor, PPC, interlaminar population correlations, and SSC) that we could not estimate reliably on a single-trial basis were not considered in the GLM analysis. Pre-target microsaccades were by far the strongest predictor of performance (*weight* = $-1.3116$; $p = 6.0757e-08$). Input layer firing rate also significantly predicted perception (*weight* = $0.3276$; $p = 0.020068$). Superficial firing rate, deep firing rate, and pupil diameter were not significant predictors (*Supplementary file 1b*, all $p > 0.5$). This indicates that, among the variables that we could estimate reliably on a single-trial basis, stable retinal images in the pre-target window are critical for behavioral performance, and elevated firing in the input layer is the most reliable physiological signature of a perceived stimulus. GLM fit parameters can be found in *Supplementary file 1c*.

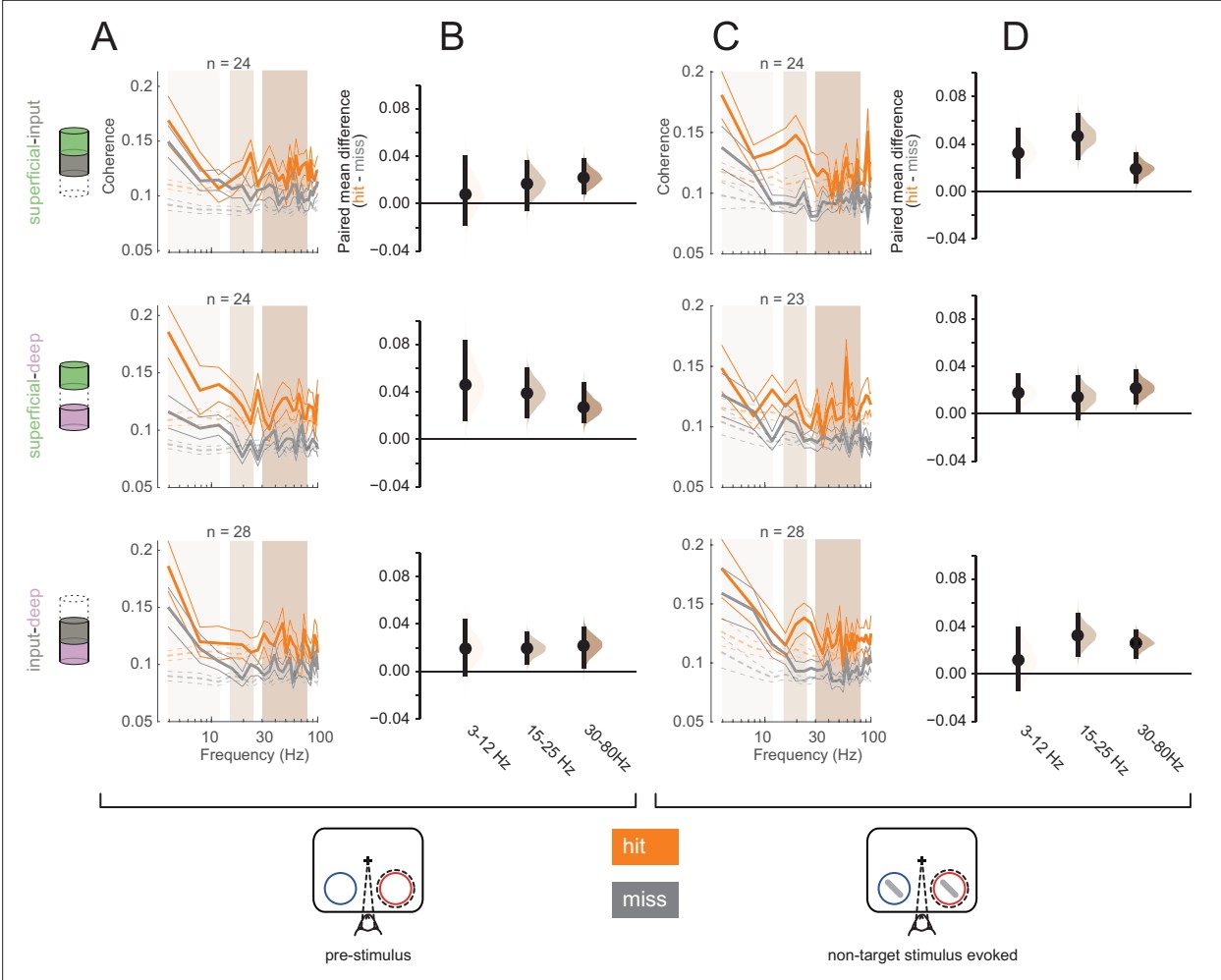

**Figure 7.** Greater interlaminar coherence in hit trials in the pre-stimulus and non-target stimulus-evoked periods. Rows correspond to different pairs of layers (top = superficial-input, middle = superficial-deep, bottom = input-deep). (**A**) Multi-unit interlaminar spike-spike coherence (SSC) calculated in the 200 ms before non-target stimulus onset in hit and miss trials (solid lines, mean ± s.e.m.). Firing rates were matched across hit and miss trials. Dashed lines represent coherence calculated with shuffled trial identities (mean ± s.e.m.). (**B**) Bootstrapped estimation plot for the paired mean difference in SSC for each pair of layers averaged over three frequency bands: 3–12 Hz, 15–25 Hz, 30–80 Hz. Curves represent the bootstrapped distribution for the paired difference, and black dots and vertical lines represent the mean and 95% confidence intervals for the paired mean difference. (**C**) Interlaminar SSC in the non-target stimulus-evoked period (60–260 ms after stimulus onset). Same conventions as in **A**. (**D**) Bootstrapped estimation plot for the paired mean difference in SSC for each pair of layers averaged over three frequency bands. Same conventions as in **B**.

The online version of this article includes the following figure supplement(s) for figure 7:

**Figure supplement 1.** Single animal interlaminar coherence.

## Discussion

We investigated the physiological processes responsible for variable behavioral outcomes at perceptual threshold. Controlling for *both* the attentive instruction (thus minimizing large-scale attentional effects) and the stimulus condition that elicited performance at a threshold level allowed us to examine the physiological and neural correlates that underlie correct vs incorrect behavioral outcomes. While this study cannot disentangle the independent roles of behavioral state fluctuations and neural fluctuations in determining behavioral outcomes, evidence suggests that differences in both are associated with hits. We found multiple lines of evidence which suggest that a state of higher arousal and eye position stability and the accompanying enhanced processing of visual stimuli contributes to accurate perception in hit trials (*Figure 8*).

Pupil diameter is elevated in hit trials (*Figure 2A–C*; *Figure 8A*), and prior studies have shown that pupil diameter is strongly linked to arousal and alertness (*Beatty and Lucero-Wagoner, 2000*;

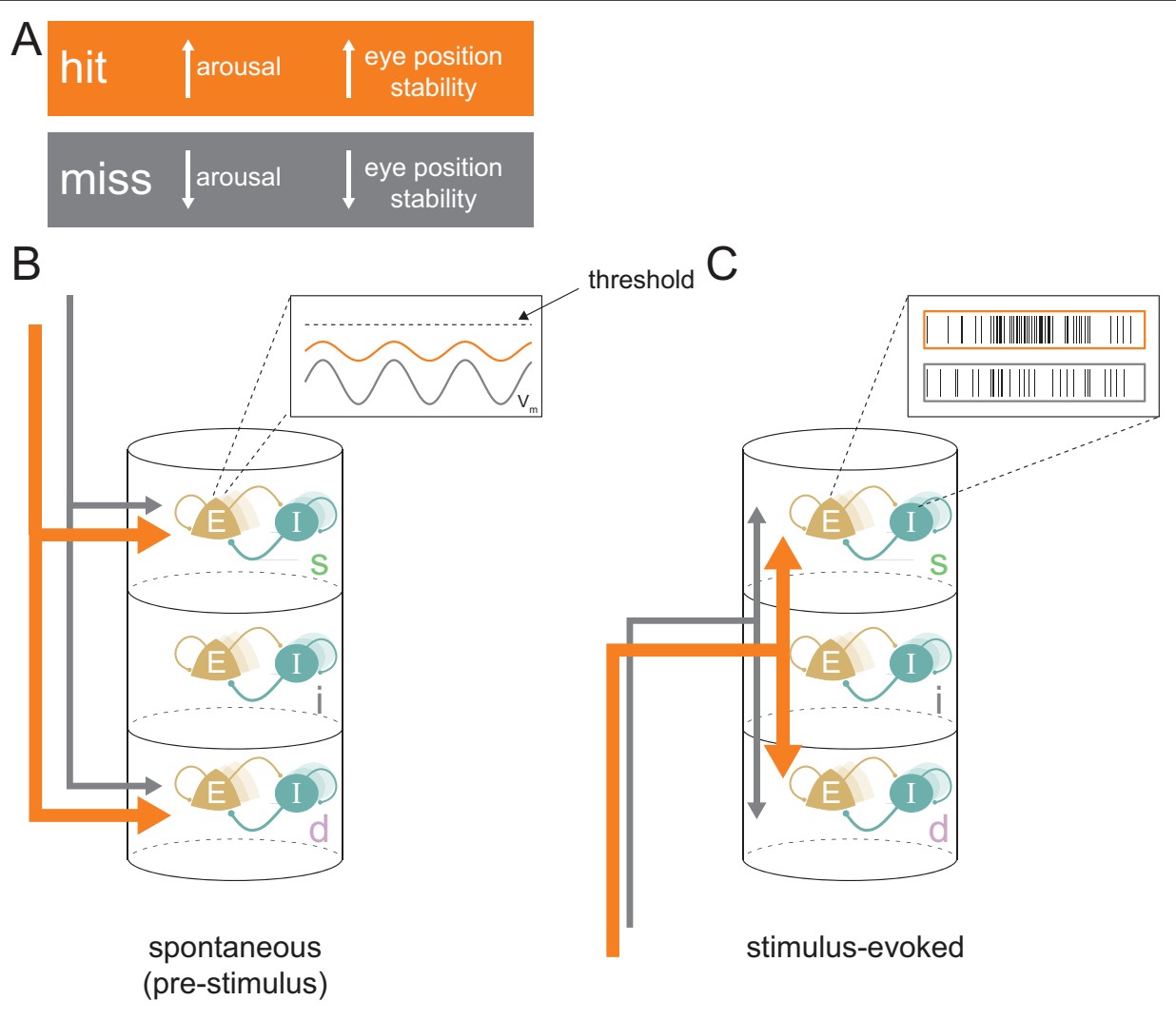

**Figure 8.** Conceptual model for stimulus processing at perceptual threshold. (**A**) Hit trials have a larger pupil diameter and fewer pre-target microsaccades, reflecting a state of increased arousal and greater eye position stability. Conversely, miss trials show decreased arousal and eye position stability. (**B**) In the spontaneous pre-stimulus period, hits are characterized by decreased variability in superficial layer broad-spiking neurons, which we hypothesize is reflective of lower membrane potential ($V_m$) variability (inset). Hit trials are also characterized by greater synchrony between the superficial and deep layers (indicated by thicker arrows), which could be reflecting a stronger top-down influence on the cortical column. (**C**) In the stimulus-evoked period there is greater interlaminar synchrony between pairs that include the input layer (represented by thicker arrows), which we propose reflects improved feed-forward propagation of information. We propose these state differences in hits contribute to elevated firing rates in response to target stimuli, particularly in the superficial layers (inset), resulting in a higher-fidelity output to downstream areas. E=excitatory; I=inhibitory; s=superficial; i=input; d=deep.

*Hess and Polt, 1964*; *Tang and Higley, 2020*). This provides evidence that a state of higher arousal may contribute to improved sensory processing. The much lower hit rate in trials with a microsaccade preceding the target (*Figure 2D*; *Figure 8A*) and our GLM analysis show that stability of retinal images is critical for accurate discrimination at threshold. It is unlikely that these two measures are reflecting the same phenomenon, as there is a very weak correlation between them over the course of a trial (*Figure 2—figure supplement 1B*).

There is a strong link between oculomotor control and attentional deployment (*Moore and Zirnsak, 2017*; *Schafer and Moore, 2011*; *Moore and Fallah, 2001*). In this study, hits and misses differ in their behavioral responses, with hit trials being characterized by a saccade to the target stimulus. Almost all of our neural results reflect differences around the time of non-target stimulus presentations during which the monkeys maintained fixation at the center of the screen and, therefore, were hundreds of

milliseconds prior to saccade planning and execution in the case of hit trials. Trials in which saccades were made to non-target stimuli were excluded from analysis, as were trials in which the monkey made a saccade to the target too soon after its presentation to have been a behavioral response to stimulus perception (see Materials and methods). The analysis of microsaccade occurrence focused on the window just before target stimulus presentation and before monkeys could begin oculomotor planning. Only the analysis of neural responses to target stimuli appears in conjunction with divergent oculomotor behavior between the hit (saccade) and miss (no saccade) trials. However, here too firing rates diverge much earlier, particularly in the input layer, than would be consistent with the effects of saccade planning (*Figure 3—figure supplement 1B*; *Steinmetz and Moore, 2014*).

Non-target stimulus contrasts were slightly different between hits and misses (mean: 33.1% in hits, 34.0% in misses, permutation test, $p = 0.02$), but the contrast of the target was higher in hits compared to misses (mean: 38.7% in hits, 27.7% in misses, permutation test, $p = 1.6 e - 31$). To control for potential effects of stimulus contrast, firing rates were first normalized by contrast before performing the analyses reported in *Figure 3*. For all other results, we considered only non-target stimuli, which had very minor differences in contrast (<1%) across hits and misses. In fact, this minor difference was in the opposite direction of our results with mean contrast being slightly higher for misses. While we cannot completely rule out any other effects of stimulus contrast, the normalization in *Figure 3* and minor differences for non-target stimuli should minimize them.

A body of evidence (see *Martinez-Conde et al., 2013*, for review) suggests that microsaccades directed toward a target stimulus reflect attention-related processing and performance (*Lowet et al., 2018*; *Engbert and Kliegl, 2003*; *Hafed and Clark, 2002*; *Gowen et al., 2007*; *Galfano et al., 2004*; *Pastukhov and Braun, 2010*; *Yu et al., 2022*). In our dataset, during the pre-target period, microsaccades toward the attended stimulus were overrepresented in correct trials (*Figure 2—figure supplement 1A*, upper left). Conversely, microsaccades toward the attended stimulus were underrepresented in incorrect trials (*Figure 2—figure supplement 1A*, lower left). Microsaccades directed toward the location of the eventual target may reflect elevated attentional deployment that can compensate for reduced sensitivity due to a higher incidence of microsaccades.

Our electrophysiological findings and their laminar patterns associated with hit trials *within* a cued attention state mirror several previous findings that are associated with the deployment of covert spatial attention. Attention has long been known to increase firing rates in V4 (*McAdams and Maunsell, 1999*; *Mitchell et al., 2007*; *Spitzer et al., 1988*), and there is evidence that this increase occurs in all cortical layers in V4 (*Nandy et al., 2017*). We find improved target vs non-target discriminability in hits (*Figure 3A*) across all cortical layers. Additionally, elevated target-evoked firing rates in hits occur across all layers in conjunction with elevated arousal (*Figure 3B–D*; *Figure 8C*). Attention reduces the variability in the firing of V4 neurons, and this reduction is thought to contribute to the improved information coding capacity of a population of neurons (*Cohen and Maunsell, 2009*; *Mitchell et al., 2007*; *Mitchell et al., 2009*; *Nandy et al., 2017*; *Moreno-Bote et al., 2014*). The reduction in Fano factor among broad-spiking superficial-layer neurons in hit trials mirrors the effects of attention (*Figure 4*). Multiple lines of evidence suggest broad- and narrow-spiking correspond to putative excitatory and inhibitory neurons respectively. Narrow-spiking neurons exhibit higher firing rates, which corresponds well with inhibitory interneuron (*Connors and Gutnick, 1990*; *Nandy et al., 2017*; *McCormick et al., 1985*; *Contreras and Palmer, 2003*; *Foehring et al., 1991*; *Povysheva et al., 2006*). Repolarization times in broad-spiking neurons are also longer, as they are in excitatory pyramidal neurons (*McCormick et al., 1985*; *Nowak et al., 2003*; *Hasenstaub et al., 2005*). Since these neurons are putative projection neurons to downstream cortical areas, this reduction in Fano factor may indicate increased reliability in stimulus encoding that could contribute to hits. Our finding is also in agreement with previous reports of higher variability in representations of unperceived stimuli in humans (*Schurger et al., 2010*). Synchronous neural activity appears to modulate perceptual and cognitive ability in a variety of contexts (*Abbas et al., 2018*; *Fries et al., 2001*; *Rohenkohl et al., 2018*; *Worden et al., 2000*). We found that deep-layer neurons exhibit less low-frequency phase-locking in hit trials (*Figure 5*). This is consistent with prior studies that find an attention-mediated reduction in the power spectrum of the spike-triggered-averaged LFP (*Fries et al., 2001*).

In examining interlaminar population synchrony, we found that hit trials were characterized by stronger feed-forward interactions across the cortical column (*Figure 6*). This state of improved interlaminar information flow could be a result of neuromodulatory or top-down processes that maintain

the cortex in a state of sustained depolarization corresponding to a state of higher arousal during hits (*McCormick et al., 2020*; *McGinley et al., 2015*). Our examination of interlaminar synchrony revealed two interesting and complementary patterns: hits were associated with greater coherence between the superficial and deep layers during spontaneous activity in the pre-stimulus period (*Figure 7A and B*; *Figure 8B*); in contrast, we found enhanced coherence between the input layer and both the output layers (superficial and deep) in the stimulus-evoked period during hits (*Figure 7C and D*; *Figure 8C*). Increased superficial-deep coherence in the pre-stimulus period could be the result of the same neuromodulatory or top-down processes. Increased synchrony between the input layer and the output layers during the stimulus-evoked period provides further evidence of stronger information propagation through the cortical circuit, and hence with improved stimulus detection (*Marshel et al., 2019*). In contrast to broad global synchrony or local correlated fluctuations, which may signal a default state of minimal processing or decreased information coding capacity (*Mitchell et al., 2009*; *Steriade et al., 1993*; *Krosigk von et al., 1993*; *Zohary et al., 1994*), these patterns of interlaminar coherence that we found suggest that successful perception at threshold is mediated by pathway-specific modulation of information flow through the laminar cortical circuit.

Prior studies showing decreased correlations under attention typically do not contain laminar information (*Cohen and Maunsell, 2009*; *Mitchell et al., 2009*) or only consider decreased correlations within a layer (*Nandy et al., 2017*). In contrast, the correlation and synchrony analyses presented here are interlaminar, which we expect could reflect improved information processing in a column, similar to principles of communication across areas (*Semedo et al., 2022*).

Taken together, our results provide insight about how information about a threshold stimulus may successfully propagate through a cortical column and influence sensory perception. Lower baseline variability among broad-spiking superficial layer neurons and decreased low-frequency synchronous activity in the deep layers could be indicative of improved capacity to encode sensory information. Higher target-evoked firing rates and elevated interlaminar synchrony could enhance the propagation of this encoded signal. These results associate pre-stimulus baseline state differences with enhanced cortical processing in the stimulus-evoked period.

Several studies have examined how information flow differs for perceived and unperceived stimuli at a more macroscopic scale (*Herman et al., 2019*; *van Vugt et al., 2018*). *van Vugt et al., 2018*, recorded from three brain regions, V1, V4, and dorsolateral prefrontal cortex, while a monkey performed a stimulus detection task at threshold. Their work supports the model that feed-forward propagation of sensory information from the visual cortex to the PFC causes a non-linear 'ignition' of association areas resulting in conscious perception (*Dehaene and Changeux, 2011*). *Herman et al., 2019*, found that conscious human perception triggers a wave of activity propagation from occipital to frontal cortex while switching off default mode and other networks. Our study provides insight into the functions of the cortical microcircuit at the columnar level that could reflect these large-scale sweeping activity changes in perception.

Overall, we identified substantial layer-specific differences in cortical activity between hits and misses at perceptual threshold, leading to the following conceptual model (*Figure 8*). During spontaneous activity, the state of elevated arousal and eye position stability during hits (*Figure 8A*) is manifested by increased interlaminar synchrony between the superficial and deep layers (*Figure 8B*, thicker orange arrows), which we propose is due to top-down influences. We predict that decreased firing variability in broad-spiking neurons in the superficial layer is caused by a lower variability in membrane potential closer to the action potential threshold among these neurons (*Figure 8B*, inset). Elevated feed-forward propagation in the stimulus-evoked period (*Figure 8C*) and a membrane potential closer to action potential threshold could both contribute to higher firing rates in the output layers of the cortex (*Figure 8C*, inset), and are indicative of greater fidelity of stimulus processing in hits. These physiological differences in the laminar microcircuit likely contribute to successful perceptual discrimination at threshold.

# Materials and methods

## Key resources table

| Reagent type (species) or resource | Designation | Source or reference | Identifiers | Additional information |
|---|---|---|---|---|
| Other | Monkey | This paper | | Species (*Macaca mulatta*) |
| Software, algorithm | MATLAB | MathWorks | | R2019a |
| Software, algorithm | Cortex | NIMH | | http://www.cortex.salk.edu/ |

## Surgical procedures

Surgical procedures have been described in detail previously (*Nandy et al., 2017*; *Nassi et al., 2015*; *Ruiz et al., 2013*). In brief, an MRI-compatible low-profile titanium chamber was placed over the prelunate gyrus, on the basis of preoperative MRI in two rhesus macaques (right hemisphere in Monkey A, left hemisphere in Monkey C). The native dura mater was then removed, and a silicone-based optically clear artificial dura (AD) was inserted, resulting in an optical window over dorsal V4 (*Figure 1—figure supplement 1A and B*). Antibiotic (amikacin or gentamicin) soaked gauze was placed in the chamber between recording sessions to prevent bacterial growth. All procedures were approved by the Institutional Animal Care and Use Committee and conformed to NIH guidelines (Salk Institute protocol number 14-00014).

## Electrophysiology

At the beginning of each recording session a plastic insert, with an opening for targeting electrodes, was lowered into the chamber and secured. This served to stabilize the recording site against cardiac pulsations. Neurons were recorded from cortical columns in dorsal V4 using 16-channel linear array electrodes ('laminar probes', Plexon Inc, Plexon V-probe). The laminar probes were mounted on adjustable X-Y stages attached to the recording chamber and positioned over the center of the prelunate gyrus under visual guidance through a microscope (Zeiss Inc, *Figure 1—figure supplement 1C*). This ensured that the probes were maximally perpendicular to the surface of the cortex and thus had the best possible trajectory to make a perpendicular penetration down a cortical column. Across recording sessions, the probes were positioned over different sites along the center of the gyrus in the parafoveal region of V4 with RF eccentricities between 2° and 7° of visual angle. Care was taken to target cortical sites with no surface microvasculature, with surface microvasculature used as reference so that the same cortical site was not targeted across recording sessions. The probes were advanced using a hydraulic microdrive (Narishige Inc) to first penetrate the AD and then through the cortex under microscopic visual guidance. Probes were advanced until the point that the top-most electrode (toward the pial surface) registered LFP signals. At this point, the probe was retracted by about 100–200 µm to ease the dimpling of the cortex due to the penetration. This procedure greatly increased the stability of the recordings and increased the neuronal yield in the superficial electrodes.

The distance from the tip of the probes to the first electrode contact was either 300 or 700 µm. The inter-electrode distance was 150 µm, thus minimizing the possibility of recording the same neural spikes in adjacent recording channels. Electrical signals were recorded extracellularly from each channel. These were then amplified, digitized, and filtered either between 0.5 Hz and 2.2 kHz (LFPs) or between 250 Hz and 8 kHz (spikes) and stored using the Multichannel Acquisition Processor system (MAP system, Plexon Inc). Spikes and LFPs were sampled at 40 and 10 kHz, respectively. LFP signals were further low-pass filtered with a sixth-order Butterworth filter with 300 Hz cut-off and downsampled to 1 kHz for further analysis. Spikes were classified as either multi-unit clusters or isolated single units using the Plexon Offline Sorter software program. Single units were identified based on two criteria: (1) if they formed an identifiable cluster, separated from noise and other units, when projected into the principal components of waveforms recorded on that electrode and (2) if the inter-spike interval distribution had a well-defined refractory period. Single units were classified as either narrow-spiking (putative interneurons) or broad-spiking (putative pyramidal cells) based on methods described in detail previously (*Mitchell et al., 2007*; *Nandy et al., 2017*). Specifically, only units with waveforms having a clearly defined peak *preceded* by a trough were potential candidates. The distribution of trough-to-peak duration was clearly bimodal (Hartigan's Dip Test, $p = 0.012$) (*Hartigan and*

*Hartigan, 1985*). Units with trough-to-peak duration less than 225 µs were classified as narrow-spiking units; units with trough-to-peak duration greater than 225 µs were classified as broad-spiking units (*Figure 1—figure supplement 1D*; teal = narrow, gold = broad).

Data was collected over 32 sessions (23 sessions in Monkey A, 9 in Monkey C), yielding a total of 413 single units (146 narrow-spiking, 267 broad-spiking) and 296 multi-unit clusters. Per session unit yield was considerably higher in Monkey C compared to Monkey A, resulting in a roughly equal contribution of both monkeys toward the population data.

## Task, stimuli, and inclusion criteria

Stimuli were presented on a computer monitor placed 57 cm from the eye. Eye position was continuously monitored with an infrared eye tracking system (ISCAN ETL-200). Trials were aborted if eye position deviated more than 1° (degree of visual angle ['dva']) from fixation. Experimental control was handled by NIMH Cortex software (http://www.cortex.salk.edu/). Eye position (all sessions) and pupil diameter (18/32 sessions) data were concurrently recorded and stored using the MAP system.

### Receptive field mapping

At the beginning of each recording session, neuronal RFs were mapped using subspace reverse correlation in which Gabor (eight orientations, 80% luminance contrast, spatial frequency 1.2 cycles/°, Gaussian half-width 2°) or ring stimuli (80% luminance contrast) appeared at 60 Hz while the monkeys maintained fixation. Each stimulus appeared at a random location selected from an 11×11 grid with 1° spacing in the appropriate visual quadrant. Spatial receptive maps were obtained by applying reverse correlation to the evoked LFP signal at each recording site. For each spatial location in the 11×11 grid, we calculated the time-averaged power in the stimulus-evoked LFP (0–200 ms after each stimulus flash) at each recording site. The resulting spatial map of LFP power was taken as the spatial RF at the recording site. For the purpose of visualization, the spatial RF maps were smoothed using spline interpolation and displayed as stacked contours plots of the smoothed maps (*Figure 1—figure supplement 1G*). All RFs were in the lower visual quadrant (lower-left in Monkey A, lower-right in Monkey C) and with eccentricities between 2 and 7 dva.

### CSD mapping

In order to estimate the laminar identity of each recording channel, we used a CSD mapping procedure (*Mitzdorf, 1985*). Monkeys maintained fixation while 100% luminance contrast ring stimuli were flashed (30 ms) centered at the estimated RF overlap region across all channels. The size of the ring was scaled to about three-quarters of the estimated diameter of the RF. CSD was calculated as the second spatial derivative of the flash-triggered LFPs (*Figure 1—figure supplement 1E*). The resulting time-varying traces of current across the cortical layers can be visualized as CSD maps (*Figure 1—figure supplement 1F*; maps have been spatially smoothed with a Gaussian kernel for aid in visualization). Red regions depict current sinks in the corresponding region of the cortical laminae; blue regions depict current sources. The input layer (Layer 4) was identified as the first current sink followed by a reversal to current source. The superficial (Layers 1–3) and deep (Layers 5–6) layers had opposite sink-source patterns. LFPs and spikes from the corresponding recording channels were then assigned to one of three layers: superficial, input, or deep.

### Attention task

In the main experiment, monkeys had to perform an attention-demanding orientation change detection task (*Figure 1A*). While the monkey maintained fixation, two achromatic Gabor stimuli (orientation optimized per recording session, spatial frequency 1.2 cycles/°, Gaussian half-width 2°, 6 contrasts randomly chosen from an uniform distribution of luminance contrasts, Monkey A: $contrast = \begin{bmatrix} 10, 18, 26, 34, 42, 50\% \end{bmatrix}$, Monkey C: $contrast\,(8\,sessions) = \begin{bmatrix} 20, 28, 36, 44, 52, 60\% \end{bmatrix}$ or $contrast\,(1\,session) = \begin{bmatrix} 30, 40, 50, 60, 70, 80\% \end{bmatrix}$) were flashed on for 200 ms and off for a variable period chosen from a uniform distribution between 200 and 400 ms. One of the Gabor's was flashed at the RF overlap region, the other at a location of equal eccentricity across the vertical meridian. The range of stimulus contrasts was the same at both locations. At the beginning of a block of trials, the monkey was spatially cued ('instruction trials') to covertly attend to one of these two spatial locations. During these instruction trials, the stimuli were only flashed at the spatially cued location. No further

spatial cue was presented during the rest of the trials in a block. At an unpredictable time drawn from a truncated exponential distribution (minimum 1 s, maximum 5 s, mean 3 s), one of the two stimuli changed in orientation. The monkey was rewarded for making a saccade to the location of orientation change. The monkey was rewarded for only those saccades where the saccade onset time was within a window of 100–400 ms after the onset of the orientation change. The orientation change occurred at the cued location with 95% probability and at the uncued location with 5% probability ('foil trials'). We controlled task difficulty by varying the degree of orientation change ($\Delta_{ori}$), which was randomly chosen from one of the following: 1°, 2°, 3°, 4°, 6°, 8°, 10°, and 12°. The orientation change in the foil trials was fixed at 4°. These foil trials allowed us to assess the extent to which the monkey was using the spatial cue, with the expectation that there would be an impairment in performance and slower reaction times compared to the case in which the change occurred at the cued location. If no change occurred before 5 s, the monkey was rewarded for maintaining fixation ('catch trials', 13% of trials). We refer to all stimuli at the baseline orientation as 'non-targets' and the stimulus flash with the orientation change as the 'target'. Monkeys initiated a median of 905 trials (range of 651–1086).

## Inclusion criteria

Of the 413 single units, we included only a subset of neurons that were visually responsive for further analysis. For each neuron we calculated its baseline firing-rate for each attention condition (attend into RF ['attend-in' or 'IN'], attend away from RF ['attend-away' or 'AWAY']) from a 200 ms window before a stimulus flash. We also calculated the neuron's contrast response function for both attention conditions (*Figure 1—figure supplement 1H*). This was calculated as the firing rate over a window between 60 and 200 ms after stimulus onset and averaged across all stimulus flashes (restricted to non-targets) of a particular contrast separately for each attention condition. A neuron was considered visually responsive if any part of the contrast response curves exceeded the baseline rate by four standard deviations for both attention conditions. This left us with 274 single units (84 narrow-spiking, 190 broad-spiking) and 217 multi-unit clusters for further analysis.

## Data analysis

### Behavioral analysis

For each orientation change condition $\Delta_{ori}$, we calculated the hit rate as the ratio of the number of trials in which the monkey correctly identified the target by making a saccadic eye movement to the location of the target over the number of trials in which the target was presented. The hit rate as a function of $\Delta_{ori}$, yields a behavioral psychometric function (*Figure 1B*). We performed this analysis independently for each recording day for each monkey, yielding a similar but distinct psychometric function for every session. Psychometric functions were fitted with a smooth logistic function (*Prins and Kingdom, 2018*). Error bars were obtained by a jackknife procedure (20 jackknives, 5% of trials left out for each jackknife). Performance for the foil trials were calculated similarly as the hit rate for trials in which the orientation change occurred at the uncued location (*Figure 1B*, square symbol). When fitting the psychometric function, we did not include the contrast of the target stimulus as a variable, and fits were calculated by including trials with target stimuli of all tested contrasts. For each fitted psychometric function in both the attend-in and attend-away conditions, we calculated the threshold of the fitted logistic function (i.e. the $\Delta_{ori}$ at which performance was mid-way between the lower and upper asymptotes). Because the threshold of the fitted function always lies somewhere on the axis of $\Delta_{ori}$, but not exactly at an orientation change presented to the subject, we then defined the threshold condition as the subset of trials in which the orientation change of the target stimulus was closest to the threshold of the fitted function (*Figure 1B*). We restricted further analysis to this threshold condition. For this threshold condition we identified the trials in which the monkey correctly identified the target as 'hit' trials and those in which the monkey failed to identify the target as 'miss' trials. Analysis of behavior, pupil diameter, and microsaccades was conducted on both the attend-in and attend-away conditions; all electrophysiological analysis was applied only to the attend-in condition.

To compare the effect of target timing across hits and misses, we determined the time between trial initiation and target presentation in all trials in the threshold condition. For comparison purposes, the contrast values of all target and non-target stimuli presented in the threshold condition were compared in hit and miss trials using a permutation test.

## Pupil diameter

The raw pupil diameter measurements from the infrared eye-tracking system could differ across days due to external factors such as display monitor illumination. To control for this, we normalized the raw data by a z-score procedure separately for each session (using the mean and standard deviation of all measurements during the session). We analyzed normalized pupil diameter traces for hit and miss trials in the threshold condition, over a time window from 100 ms before to 100 ms after all stimulus presentations (non-target and target), excluding the first stimulus presentation in a trial. The first stimulus was excluded to avoid pupil diameter changes due to the pupillary near response caused by acquiring fixation (*McDougal and Gamlin, 2015*). The pupil diameter was averaged over this time period and compared across conditions using bootstrap estimation and t-test. Distribution violin plots were generated using kernel density estimation (*Hoffmann, 2015*) (bandwidth(hit)=0.0801, bandwidth(miss)=0.0648). When analyzing pupil diameter, we did not include the contrast of the target and non-target stimuli as a variable, and all analyses were performed by including stimuli of all tested contrasts.

## Microsaccade analysis

Saccadic eye movements were detected using ClusterFix (*König and Buffalo, 2014*). We identified microsaccades by filtering for eye movements with amplitudes between 0.1 and 1 dva. We then split all trials in the threshold condition into two groups: those in which a microsaccade was detected in the 400 ms preceding the target stimulus presentation, and those without a detected microsaccade. We calculated the hit rate for trials within those two groups. For all trials in which a target stimulus was presented at the attended location, we determined the direction of all microsaccades in the 400 ms period preceding target presentation, relative to both the attended and unattended stimuli. The relative microsaccade direction was defined as the angle between two vectors: the one defined by the eye positions at the beginning and end of the microsaccade, and the vector from the initial eye position to the center of the stimulus (calculated separately for attended and unattended stimuli). Relative microsaccade directions were grouped into 12 bins from 0° to 360°. The distribution of relative microsaccade directions were calculated separately for correct and incorrect trials, relative to both the attended and unattended stimuli (*Figure 2—figure supplement 1A*).

We next created a null distribution of relative microsaccade direction. This was done by pooling together microsaccades from correct and incorrect trials and then sampling with replacement from this pooled data (bootstrap procedure [*Efron and Tibshirani, 1993*]; 1000 samples). The number of microsaccades chosen for each sample was the same as the number in correct or incorrect trials, respectively. These bootstrapped samples were used to create 99.5% confidence intervals for the count of microsaccades expected in each of the 12 bins. A bin was considered significantly different from chance if it's true count fell outside this confidence interval.

We calculated microsaccade rate for an entire trial by dividing the total number of detected microsaccades in the whole trial by the trial length (4592 total trials). The Pearson correlation between microsaccade rate and mean normalized pupil diameter (see above) for the trial was calculated for all trials with pupil diameter data, regardless of trial type or outcome (*Figure 2—figure supplement 1B*). Not pictured in *Figure 2—figure supplement 1B* but included in correlation analysis were trials with a mean normalized pupil diameter greater than 2 or less than –2 (~4% of trials). Only four of these trials were longer than 1 s, out of which two trials contained detected microsaccades. The mean pupil diameter in these trials is shown in *Figure 2—figure supplement 1B*, inset. Inter-microsaccade time was calculated as the time between microsaccade onset of microsaccades detected in the same trial. 4% of microsaccades separated by >538 ms are excluded from *Figure 2—figure supplement 1D* as they were more than 1.5× the interquartile range above the third quartile. When analyzing microsaccades, we did not include the contrast of the target and non-target stimuli as a variable, and all analyses were performed by including stimuli of all tested contrasts.

## Decoding analysis

For each single- or multi-unit neuron, we extracted spike counts from 60 to 260 ms following all non-target or target stimulus onsets in the threshold condition. Using these spike counts, we fit a Poisson distribution to estimate the mean firing rate for each neuron in each of four stimulus conditions (non-targets and targets in hit and miss trials). Responses to stimuli of all contrasts were pooled together to

create the Poisson fit. We then created a pseudo-population of neurons in each layer. We generated spike counts drawn from the fitted Poisson distributions to create synthetic spike counts for target and non-target stimuli in hits and misses separately (1000 repeats). We used linear discriminant analysis to decode target from non-target stimuli. Decoders were trained separately for hit and miss trials. The procedure was repeated for a 20-fold cross-validation. We calculated the chance performance by training the decoder on data generated from all trials in the threshold condition (both hits and misses) and with shuffled labels (target or non-target).

## Firing rate

Firing rates were normalized per neuron to that neuron's maximum stimulus-evoked response to each contrast before being combined across contrasts and trial types. PSTH of firing rates were calculated in 30 ms bins shifted in 5 ms increments. For each tested contrast level, we calculated each neuron's maximum average firing rate in any bin from 0 to 260 ms after stimulus onset in hit and miss trials together. We divided all bins in the entire PSTH for that contrast by this maximum firing rate to obtain the normalized firing rate for each contrast. We then combined PSTHs across contrasts and split the data between hit and miss trials. We averaged stimulus-evoked firing rates from 60 to 260 ms following non-target or target stimulus presentations. We used bootstrapped estimation to compare firing rates in hit and miss trials in a paired comparison. This was done for all single- and multi-unit clusters, as well as broad- and narrow-spiking single units in each layer. Firing rates were also compared across hit and miss trials by paired t-test for each group. To calculate the time of firing rate divergence, we compared the difference in each single neuron or multi-unit firing rate in the two conditions over time. At each time point, we performed a Wilcoxon rank sum test comparing the firing rates across hits and misses, and defined the divergence point as the first time the firing rates were significantly different. Divergence was calculated separately for each layer. To determine the time at which firing rates rose above baseline levels, we used bootstrapping to estimate 95% confidence intervals for each neuron's pre-stimulus firing rate (0–100 ms before target or non-target stimulus onset). We then calculated the target PSTH for each neuron in 30 ms bins shifted in 5 ms increments. We defined the response latency as the first time bin in which the neuron's firing rate in the PSTH exceeded the upper limit of the 95% confidence interval of baseline firing. We calculated the response latency independently for hit and miss trials for each neuron.

## Fano factor

Trial-to-trial variability was estimated by the Fano factor, which is the ratio of the variance of the spike counts across trials over the mean of the spike counts for each broad- and narrow-spiking single unit. The Fano factor was calculated over non-overlapping 20 ms time bins in a window from 200 ms prior to each non-target flash onset to 200 ms after each non-target flash onset for hit and miss trials in the threshold condition. To compare across conditions, we calculated the Fano factor modulation index (MI), defined as

$$\text{MI} = \frac{FF_{hit} - FF_{miss}}{FF_{hit} + FF_{miss}}$$

where $FF_{hit}$ and $FF_{miss}$ represent the Fano factor for a given unit in hit and miss trials respectively at each point in time with respect to non-target stimulus onset. The Fano factor MI was averaged from 0 to 60 ms prior to non-target stimulus onset (disregarding stimulus contrast) and compared across trial types in the threshold condition for each sub-population.

## Pairwise phase consistency

We calculated PPC (*Vinck et al., 2010*) for single and multi-units in the non-target pre-stimulus period (0–200 ms preceding onset regardless of stimulus contrast) in trials in the threshold condition. Although PPC is unbiased by spike count, we set a threshold of 50 spikes for analysis so that only units with enough spikes for a reliable estimate of PPC were included (superficial: $n = 26$, input: $n = 41$, deep: $n = 64$). LFP phase was calculated using Morlet wavelets. PPC for each unit was calculated for the phase of the LFP recorded on the same channel and averaged in three frequency bands (3–12 Hz, 15–25 Hz, and 30–80 Hz). PPC was calculated separately for hit and miss trials and compared across trial outcomes by t-test, corrected for multiple comparisons.

## Canonical correlation analysis

We used CCA (*Semedo et al., 2022*; *Hotelling, 1992*) to capture the correlation between layers at different time periods and with different amounts of temporal delays. We considered all possible combinations of pairs of layers. We took two windows of activity, one in each layer, in either the pre-stimulus period (0–200 ms before non-target stimulus onset) or stimulus-evoked period (60–260 ms after non-target stimulus onset), including responses to stimuli of all contrasts. Window length was 50 ms and the window was advanced in 10 ms steps. The activity within each window was then binned using 10 ms bins. We reported correlation associated with the first two canonical pairs, and calculated it separately for hit and miss trials. For each pair of layers, we limited CCA to sessions in which there were at least two neurons recorded in both layers. Feed-forward (FF) layer pairs were defined as follows: input to superficial, input to deep, and superficial to deep. The correlations along the FF signaling pathways ($C_{FF}$) were calculated as the mean correlation at positive delays:

$$C_{FF} = \frac{\sum\limits_{dt>0} C\left(t, dt\right)}{N_{dt>0}}$$

where t is the time following response onset, $dt$ is the interlaminar delay involved between windows from two layers, $C(t, dt)$ is the corresponding correlation. $N_{dt>0}$ is the number of positive delays investigated. The values in *Figure 6B* correspond to $C_{FF}$ in each session.

## Spike-spike coherence

For each recording session, all spikes recorded from visually responsive single and multi-units in each layer were combined into a single spike train for that layer (layer multi-unit). Separately for both the pre-stimulus and non-target stimulus-evoked periods, we randomly deleted spikes from the layer multi-unit with a higher firing rate so that the firing rates were matched across hit and miss trials. SSC was calculated for each of the three possible pairs of layer multi-units in each session for both the pre-stimulus (0–200 ms preceding stimulus onset) and non-target stimulus evoked period (60–260 ms following non-target stimulus onset, including all stimuli regardless of contrast) separately for hit and miss trials using Chronux (NW = 1; K=1; http://chronux.org; *Mitra, 2007*; *Mitra and Pesaran, 1999*). To control for differences in firing rates across hit and miss trials we used a rate matching procedure (*Mitchell et al., 2009*). For estimation statistics, interlaminar SSC values were calculated for each frequency and subsequently averaged across three frequency bands: 3–12 Hz, 5–15 Hz, and 30–80 Hz and compared across hit and miss trials for each pair of layers in each recording session. For null hypothesis testing, we calculated the SSC modulation index, defined as

$$\text{MI} = \frac{SSC_{hit} - SSC_{miss}}{SSC_{hit} + SSC_{miss}}$$

The SSC MI was calculated for each frequency and subsequently averaged across three frequency bands: 3–12 Hz, 5–15 Hz, and 30–80 Hz. MI values for each frequency band were compared to zero by t-test, Bonferroni corrected for multiple comparisons. We tested for interaction effects with a three-factor ANOVA, with frequency, pair of layers, and time window (pre or post stimulus) as factors. We calculated a shuffled distribution of SSC by shuffling the trial identities of the spikes in one of the layers in the pair. We then calculated SSC with the shuffled trial identities. This procedure was repeated 10 times to create the shuffled distribution.

## GLM quantification

To compare how well our results can predict behavioral performance, we fit a GLM to the response of the monkeys in trials in the threshold condition (*Davis et al., 2020*). We included five regressors in our analysis: (1) average pupil diameter during the trial, (2) number of microsaccades in the pre-target window (0–400 ms before target stimulus onset), and average target-evoked multi-unit firing rate in the (3) superficial, (4) input, and (5) deep layers. We calculated the average target-evoked firing rate by averaging the firing rate of all single- and multi-units in a given layer 60–260 ms after target stimulus onset in each trial. In order to be able to compare weights across regressors, each regressor was transformed into a z-score before being included in the model. We fit the GLM

using a logit link function, using the predictors to regress the categorical binary trial outcome (hit or miss). A total of 309 trials were included in the GLM. Stimulus contrast was not a variable in the GLM.

## Acknowledgements

This research was supported by NIH R01 EY021827 to JHR and ASN, NIH R01 EY032555, NARSAD Young Investigator Grant, Ziegler Foundation Grant, Yale Orthwein Scholar Funds & Lawrence Family Young Investigator Funds to ASN, NIH R01 EY034605 and NIH R00 EY025026 to MPJ, and by NEI core grants for vision research P30 EY019005 to the Salk Institute and P30 EY026878 to Yale University. MM was supported by training grants T32-NS007224 and T32-NS041228 to Yale University. We would like to thank Catherine Williams and Mat LeBlanc for their excellent animal care.

## Additional information

### Funding

| Funder | Grant reference number | Author |
|---|---|---|
| National Eye Institute | R01 EY021827 | John H Reynolds<br>Anirvan S Nandy |
| National Eye Institute | R01 EY032555 | Anirvan S Nandy |
| National Eye Institute | R01 EY034605 | Monika P Jadi |
| National Eye Institute | R00 EY025026 | Monika P Jadi |
| National Institute of Neurological Disorders and Stroke | T32 NS007224 | Mitchell P Morton |
| National Institute of Neurological Disorders and Stroke | T32 NS041228 | Mitchell P Morton |
| National Eye Institute | P30 EY019005 | John H Reynolds |
| National Eye Institute | P30 EY026878 | Anirvan S Nandy |

The funders had no role in study design, data collection and interpretation, or the decision to submit the work for publication.

### Author contributions

Mitchell P Morton, Formal analysis, Writing - original draft; Sachira Denagamage, Software; Isabel J Blume, Formal analysis; John H Reynolds, Funding acquisition, Writing - review and editing; Monika P Jadi, Conceptualization; Anirvan S Nandy, Conceptualization, Supervision, Funding acquisition, Investigation, Methodology, Writing - review and editing

### Author ORCIDs

Mitchell P Morton ⓘ http://orcid.org/0000-0001-9185-1633
John H Reynolds ⓘ https://orcid.org/0000-0001-6988-4607
Monika P Jadi ⓘ https://orcid.org/0000-0003-1092-5026
Anirvan S Nandy ⓘ https://orcid.org/0000-0002-4225-5349

### Ethics

All procedures were approved by the Institutional Animal Care and Use Committee and conformed to NIH guidelines.(Salk Institute protocol number 14-414 00014).

Reviewer #1 (Public review): https://doi.org/10.7554/eLife.91722.4.sa1
Author response https://doi.org/10.7554/eLife.91722.4.sa2

# Additional files

## Supplementary files

• Supplementary file 1. Supplementary tables. (a) Corresponding null hypothesis testing results. Null hypothesis testing results corresponding to the estimation statistics-based analyses. (b) Generalized linear model (GLM) coefficient values. Coefficients and significance values of the variables used in the GLM analysis. (c) GLM summary. Additional summary statistics of the GLM analysis.

• MDAR checklist

## Data availability

This study used data from a previously published study: *Nandy et al., 2017*. Data used in this study has been deposited in Zenodo: https://zenodo.org/records/13968327.

The following dataset was generated:

| Author(s) | Year | Dataset title | Dataset URL | Database and Identifier |
|---|---|---|---|---|
| Morton M, Denagamage S, Blume I, Reynolds J, Jadi M, Nandy A | 2024 | Brain state and cortical layer-specific mechanisms underlying perception at threshold | https://doi.org/10.5281/zenodo.13968327 | Zenodo, 10.5281/zenodo.13968327 |

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
