## [Editor Report · eLife Assessment]

This **useful** study by Nandy and colleagues examined relationships between behavioral state, neural activity in cortical area V4, and trial-by-trial variability in the ability to detect weak visual stimuli. They present **solid** evidence indicating that certain changes in arousal and eye-position stability, along with patterns of synchrony in the activity of neurons in different layers of V4, can show modest correspondences to changes in the ability to correctly detect a stimulus. These findings are likely to be of interest to those who seek a deeper understanding of circuit mechanisms that underlie perception.

---

## [Referee Report · Reviewer #1 (Public review)]

Summary:

In this study, Nandy and colleagues examine neural, physiological and behavioral correlates of perceptual variability in monkeys performing a visual change detection task. They used a laminar probe to record from area V4 while two macaque monkeys detected a small change in stimulus orientation that occurred at a random time in one of two locations, focusing their analysis on stimulus conditions where the animal was equally likely to detect (hit) or not-detect (miss) a briefly presented orientation change (target). They discovered two behavioral and physiological measures that are significantly different between hit and miss trials - pupil size tends to be slightly larger on hits vs. misses, and monkeys are more likely to miss the target on trials in which they made a microsaccade shortly before target onset. They also examined multiple measures of neural activity across the cortical layers and found some measures that are significantly different between hits and misses.

Strengths:

Overall the study is well executed and the analyses are appropriate (with some possible caveats discussed below).

Weaknesses:

I have two remaining concerns. First, with the exception of the pre-target microsaccades, the correlates of perceptual variability (differences between hits and misses) appear to be weak and disconnected. The GLM analysis of the predictive power of trial outcome based on the behavioral and neural measures is only discussed at the end of the paper. This analysis shows that some of the measures have no significant predictive power, while others cannot be examined using the GLM analysis because these measures cannot be estimated in single trials. Given these weak and disconnected effects, my overall sense is that the current results provide a limited advance to our understanding of the neural basis of perceptual variability.

In addition, because the authors combine data across stimulus contrasts, I am somewhat uneasy about the possible confounding effect of contrast. As expected, stimulus contrast affected the probability of hits vs. misses. Independently, contrast may have affected some of the physiological measurements. Therefore, showing that contrast is not the source of the covariations between the physiological/behavioral measurements and perception can be challenging, and I am not convinced that the authors have ruled this out as a possible confound. It is unclear why the authors had to vary contrast in the first place, and why the analyses had to be done by combining the data across contrasts or by ignoring contrast as a variable (e.g., in the GLM analysis).

---

## [Author Response]

The following is the authors’ response to the previous reviews.

As you can see from the assessment (which is unchanged from before) and the reviews included below, the reviewers felt that the revisions did not yet address all of the major concerns. There was agreement that the strength of evidence would be upgraded to "solid" by addressing, at minimum, the following:(1) Which of the results are significant for individual monkeys; and(2) How trials from different target contrasts were analyzed

In this revision, we have addressed the two primary editorial recommendations:

(1) We apologize if this information was not clear in the previous version. We have updated Table 1 to highlight clearly the significant results for individual monkeys. Six of our key results – pupil diameter (Fig 2B), microsaccades (Fig 2D), decoding performance for narrow-spiking units (Fig 3A), decoding performance for broad-spiking units (Fig 3B), target-evoked firing rate for all units (Fig 3E) and target-evoked firing rate for broad-spiking units (Fig 3F) – are significant for individual animals and therefore gives us high confidence regarding our results. Please also note that we present all results for individual animals in the Supplementary figures accompanying each main figure.

(2) We have updated the manuscript and methods to explain how trials of each contrast were included in each analysis, and how contrast normalization was performed for the analysis in Figure 3. In addition, we discuss this point in the Discussion section, which we quote below:

“Non-target stimulus contrasts were slightly different between hits and misses (mean: 33.1% in hits, 34.0% in misses, permutation test, 𝑝 = 0.02), but the contrast of the target was higher in hits compared to misses (mean: 38.7% in hits, 27.7% in misses, permutation test, 𝑝 = 1.6 𝑒 − 31). To control for potential effects of stimulus contrast, firing rates were first normalized by contrast before performing the analyses reported in Figure 3. For all other results, we considered only non-target stimuli, which had very minor differences in contrast (<1%) across hits and misses. In fact, this minor difference was in the opposite direction of our results with mean contrast being slightly higher for misses. While we cannot completely rule out any other effects of stimulus contrast, the normalization in Figure 3 and minor differences for non-target stimuli should minimize them.”

**Reviewer #1 (Public Review):**
Summary:In this study, Nandy and colleagues examine neural, physiological and behavioral correlates of perceptual variability in monkeys performing a visual change detection task. They used a laminar probe to record from area V4 while two macaque monkeys detected a small change in stimulus orientation that occurred at a random time in one of two locations, focusing their analysis on stimulus conditions where the animal was equally likely to detect (hit) or not-detect (miss) a briefly presented orientation change (target). They discovered two behavioral and physiological measures that are significantly different between hit and miss trials - pupil size tends to be slightly larger on hits vs. misses, and monkeys are more likely to miss the target on trials in which they made a microsaccade shortly before target onset. They also examined multiple measures of neural activity across the cortical layers and found some measures that are significantly different between hits and misses.Strengths:Overall the study is well executed and the analyses are appropriate (though several issues still need to be addressed as discussed in Specific Comments).

Thank you.

Weaknesses:My main concern with this study is that, with the exception of the pre-target microsaccades, the correlates of perceptual variability (differences between hits and misses) appear to be weak, potentially unreliable and disconnected. The GLM analysis of predictive power of trial outcome based on the behavioral and neural measures is only discussed at the end of the paper. This analysis shows that some of the measures have no significant predictive power, while others cannot be examined using the GLM analysis because these measures cannot be estimated in single trials. Given these weak and disconnected effects, my overall sense is that the current results provide limited advance to our understanding of the neural basis of perceptual variability.

Please see our response above to item #1 of the editorial recommendation. Six of our key results are individually significant in both animals giving us high confidence about the reliability and strength of our results.

Regarding the reviewer’s comment about the GLM, we note (also stated in the manuscript) that among the measures that we could estimate reliably on a single trial basis, two of these – pre-target microsaccades and input-layer firing rates – were reliable signatures of stimulus perception at threshold. This analysis does not imply that the other measures – Fano Factor, PPC, inter-laminar population correlations, SSC (which are all standard tools in modern systems neuroscience, and which cannot be estimated on a single-trial basis) – are irrelevant. Our intent in including the GLM analyses was to *complement* the results reported from these across-trial measures (Figs 4-7) with the predictive power of single-trial measures.

While no study is entirely complete in itself, we have attempted to synthesize our results into a conceptual model as depicted in Fig 8.

**Reviewer #2 (Public Review):**
Strengths:The experiments were well-designed and executed with meticulous control. The analyses of both behavioural and electrophysiological data align with the standards in the field.

Thank you.

Weaknesses:Many of the findings appear to be subtle differences and incremental compared to previous literature, including the authors' own work. While incremental findings are not necessarily a problem, the manuscript lacks clear statements about the extent to which the dataset, analysis, and findings overlap with the authors' prior research. For example, one of the main findings, which suggests that V4 neurons exhibit larger visual responses in hit trials (as shown in Fig. 3), appears to have been previously reported in their 2017 paper.

We respectfully disagree with the assessment that the findings reported here are incremental over the results reported in our prior study (Nandy et al,. 2017). In the previous study, we compared the laminar profile of neural modulation due to the deployment of attention i.e. the main comparison points were the attend-in and the attend-away conditions while controlling for visual stimulation. In this study, we go one step further and home in on the attend-in condition and investigate the differences in the laminar profile of neural activity (and two additional physiological measures: pupil and microsaccades) when the animal either correctly reports or fails to report a stimulus with equal probability. We thus control for *both* the visual stimulation and the cued attention state of the animal. While there are parallels to our previous results (as the reviewer correctly noted), the results reported here cannot be trivially predicted from our previous results. Please also note that we discuss our new results in the context of prior results, from both our group and others, in the manuscript (lines 310-332).

Furthermore, the manuscript does not explore potentially interesting aspects of the dataset. For instance, the authors could have investigated instances where monkeys made 'false' reports, such as executing saccades towards visual stimuli when no orientation change occurred, which allows for a broader analysis that considers the perceptual component of neural activity over pure sensory responses. Overall, lacking broad interest with the current form.

We appreciate the reviewer’s feedback on analyzing false alarm trials. Our focus for this study was to investigate the behavioral and neural correlates accompanying a correct or incorrect perception of a target stimulus presented at perceptual threshold. False alarm trials, by definition, do not include a target presentation. Moreover, false alarm rates rapidly decline with duration into a trial, with high rates during the first non-target presentation and rates close to zero by the time of the eighth presentation (see figure). Investigating false alarms will thus involve a completely different form of analysis than we have undertaken here. We therefore feel that while analyzing false alarm trials will be an interesting avenue to pursue in the future, it is outside the scope of the present study.

**Author response image 1. sa2fig1:**